# Human Semaphorin-4A drives Th2 responses by binding to receptor ILT-4

Ning Lu[1,2], Ying Li[1], Zhiqiang Zhang [2,3], Junji Xing [3], Ying Sun[4,5], Sheng Yao[6,7] & Lieping Chen[6]

Semaphorin-4A (Sema4A) has been implicated in the co-stimulation of T cells and drives Th1 immune responses by binding to the receptor T-cell immunoglobulin and mucin domain protein 2 (Tim-2) in mice. Here we show that human, but not murine, Sema4A is preferentially expressed on antigen-presenting cells, and co-stimulates CD4[+] T-cell proliferation and drives Th2 responses. By employing two independent cloning strategies, we demonstrate that Immunoglobulin-like transcript 4 (ILT-4) is a receptor for human SEMA4A (hSEMA4A) on activated CD4[+] T cells. We also find hSEMA4A to be highly expressed in human asthmatic lung tissue, implying its potential function in disease pathogenesis. Our study defines a different biological function of hSEMA4A from its murine homolog through its binding to the receptor of ILT-4 to co-stimulate CD4[+]T cells and regulate Th2 cells differentiation.

[1] CAS Key Laboratory of Infection and Immunity, Institute of Biophysics, Chinese Academy of Sciences, Beijing 100101, China. [2] Department of Immunology and Center for Cancer Immunology Research, The University of Texas M.D. Anderson Cancer Center, Houston TX 77030, USA. [3] Immunobiology and Transplant Research, Houston Methodist Hospital and Methodist Hospital Research Institute, Texas Medical Center, Houston TX 77030, USA. [4] Division of Asthma, Allergy and Lung Biology, MRC-Asthma UK Centre for Allergic Mechanisms of Asthma, King's College London, London SE1 1YZ, UK. [5] Department of Immunology, School of Basic Medical Sciences, Capital Medical University, Beijing 100069, China. [6] Department of Immunobiology, Medicine and Dermatology, Cancer Immunology Program at Yale Cancer Center, Yale University School of Medicine, New Haven CT 06519, USA. [7] Oncology and Cellular Therapy, TopAlliance Biosciences, Rockville MD 20850, USA. Correspondence and requests for materials should be addressed to N.L. (email: ninglu@ibp.ac.cn)

Semaphorins are a large family of secreted and membrane-bound glycoproteins that were initially implicated in axon guidance and neural development[1,2], and are divided into eight subclasses. Subclasses III–VII contain vertebrate semaphorins. Class III semaphorins are secreted, classes IV–VI semaphorins are transmembrane proteins, and class VII semaphorins are membrane-associated via glycosyl phosphatidylinositol (GPI) linkage. Semaphorins have been implicated in axon outgrowth, angiogenesis, bone differentiation, cardiovascular development, and regulation of immune responses[3–5].

Semaphorin-4A (Sema4A) was originally identified in developing embryos, and its transcript levels increase gradually throughout embryonic development[6]. In addition to its expression during embryogenesis, Sema4A mRNA is detectable in adult brain, lung, kidney, testis, and spleen. In murine immune system, Sema4A is not expressed by resting T cells. Its expression can be induced on activated T cells[7]. Resting B cells express low levels of Sema4A, but activation with anti-CD40 antibody can upregulate Sema4A expression. Sema4A is preferentially expressed by dendritic cells (DCs). It can provide T-cell co-stimulation[7]. Addition of Sema4A-Fc fusion protein enhances T-cell proliferation and cytokine production after stimulation with anti-CD3 antibody. In addition, soluble Sema4A-Fc protein enhances the mixed lymphocyte reactions (MLR) between allogeneic T cells and DCs, while anti-Sema4A antibody blocks the MLR. Administration of Sema4A protein enhances the generation of antigen-specific T cells in vivo. By contrast, administration of anti-Sema4A antibody blocks antigen-specific T-cell priming[7]. In an experimental autoimmune encephalomyelitis (EAE) model, anti-Sema4A antibody treatment inhibits the development of EAE[7,8]. In another model, administration of Sema4A protein also downregulates the severity of allergic airway response in mice[9,10]. Furthermore, T cells from Sema4A-deficient mice differentiate poorly into interferon-γ (IFN-γ)-secreting Th1 cells, and Th1 responses are severely impaired suggested that Sema4A is required not only for T-cell co-stimulation but also for Th1 cell differentiation[8,11–14].

Receptors or receptor complexes that mediate semaphorin signaling include the proteins from the neuropilin and plex-infamilie[15,16], plexins (plexin A1-A4, plexin B1–3, plexin C1, and plexin D1) and neuropilins (Nrp1 and Nrp2) are the primary semaphorins receptors[17,18]. Sema4A binds to plexin D1 to suppress vascular endothelial growth factor-mediated migration and proliferation of endothelial cells, while Sema4A induces cell morphological changes through receptors plexin B1, B2, or B3[19,20]. In addition, Sema4A is required for the function and stability of regulatory T (Treg) cells by binding to neupilin-1 (Nrp1) on Treg[21–24].

T-cell immunoglobulin (Ig) and mucin domain-containing protein 2 (Tim-2), a molecule unrelated to plexins and neuropilins, was identified as a Sema4A receptor expressed on the surface of activated T cells in mice[7]. However, Sema4A-Fc fusion protein attenuates airway inflammation and Th2 immune responses even in Tim-2-deficient mice[11]. The functions of Tim-2 binding to Sema4A are still unclear. In addition, there is no human ortholog of Tim-2[25]. So far, the biological functions of Sema4A in human immune system are unknown.

Here we demonstrate that, unlike mouse Sema4A, which preferentially induces Th1 immune responses, human SEMA4A (hSEMA4A) induces robust Th2 responses. By using expression cloning from an activated human CD4+ T-cell library, and a receptor assay system, we identify immunoglobulin-like transcript 4 (ILT-4) as the receptor for hSEMA4A.

## Results

### Sema4A highly expressed in human DCs co-stimulates T cells.

To investigate the function of Sema4A in humans, we first detected the expression of Sema4A in various populations of human monocytes, DCs, T cells, granulocytes, as well as B cells, NK cells, mast cells sorted from human peripheral blood mononuclear cells (PBMC) by using microarray gene expression analysis. Sema4A was highly expressed in CD4+CD11c+ myeloid DCs (mDCs), and moderately expressed in monocyte-derived DCs, B cells, monocytes, and CD45RO+CD4+CRTH2+memory Th2 cells (Fig. 1a). Using Q-PCR detection, we directly confirmed that Sema4A was most highly expressed in mDC, followed by B cells and memory Th2 cells, but not in naive (Tn), central memory (Tcm), and effector memory (Tem) CD4+T cells (Fig. 1b). To analyze SEMA4A expression at the protein level, we generated monoclonal antibodies (mAbs) against SEMA4A by immunizing mice with the lysate of SEMA4A-expressing mouse fibroblast L cells and screening the hybridomas for reactivity to L cells expressing membrane-bound SEMA4A, but not to L cells transfected with mock vector of pMXS-GFP-Flag (Supplementary Fig. 1). By using anti-SEMA4A mAb, we analyze the SEMA4A expression on the surface of different populations of B cell and CD4+mDCs from human tonsils and CD4+ T cells from PBMC by flow cytometry analysis with anti-SEMA4A mAb (clone 70). Consistent with the expression of Sema4A mRNA, SEMA4A was highly expressed on BDCA1+CD4+DCs, IgD−CD38− Germinal center (GC) B cells and CRTH2+memory Th2 cells, and lowly expressed on IgD−CD38+memory B cells, IgD+CD38−naive B cells and CD45RO+CRTH2−CD4+ T cells. SEMA4A was not expressed on BDCA3+BDCA1−DCs and CD45RA+CD4+naive T cells (Fig. 1c). In addition, immunohistochemistry analysis revealed the expression of SEMA4A in human tonsils, where SEMA4A is majorly expressed on CD11c+DCs in the interfollicular areas, and CD11c− cells in GC, which should be B cells (Fig. 1d, e).

Because hSEMA4A is most highly expressed on DCs, we wondered whether it co-stimulates T-cell responses. The sorted human CD45RO−CD45RA+naive CD4+ T cells were labeled with Carboxyfluorescein Succinimidyl Ester (CFSE) and stimulated with a suboptimal dose of immobilized anti-CD3 mAb (OKT3) (0.2 μg ml−1) either in the presence of Sema4A-transfected or non-transfected L cells, or in the presence of immobilized SEMA4A-Fc fusion proteins or human immunoglobulin G (hIgG). Both SEMA4A-expressing L cells and SEMA4A-Fc protein strongly promoted the proliferation of activated CD4+ T cells compared with non-transfected L cells and hIgG, respectively (Fig. 2a). Inclusion of anti-SEMA4A mAb completely blocked T-cell proliferation co-stimulated by SEMA4A-transfected L cells while mouse immunoglobulin G (mIgG) had no effects (Fig. 2b). To explore the physiological role of SEMA4A in T-cell proliferation, purified peripheral blood CD11c+CD4+mDCs were cultured with CD45RO−CD45RA+CD4+naive T cells from allogeneic donors at a 1:5 ratio for 7 days in the presence of anti-SEMA4A mAb or a mIgG1 isotype control. The allogeneic mDCs stimulated robust T-cell proliferation of CD4+T cells, which was completely blocked by anti-SEMA4A mAb but not the mIgG1 (Fig. 2c). Thus, similar to mouse Sema4A, hSEMA4A is also highly expressed on DC, which co-stimulate T-cell proliferation.

### SEMA4A induces Th2 responses.

Because Sema4A has been implicated in promoting Th1 responses in mice[7,8], we investigated the role of hSEMA4A in regulating the differentiation of human CD4+ T cells. Naive CD4+T cells purified from human PBMC were cultured with hSEMA4A-expressing L cells

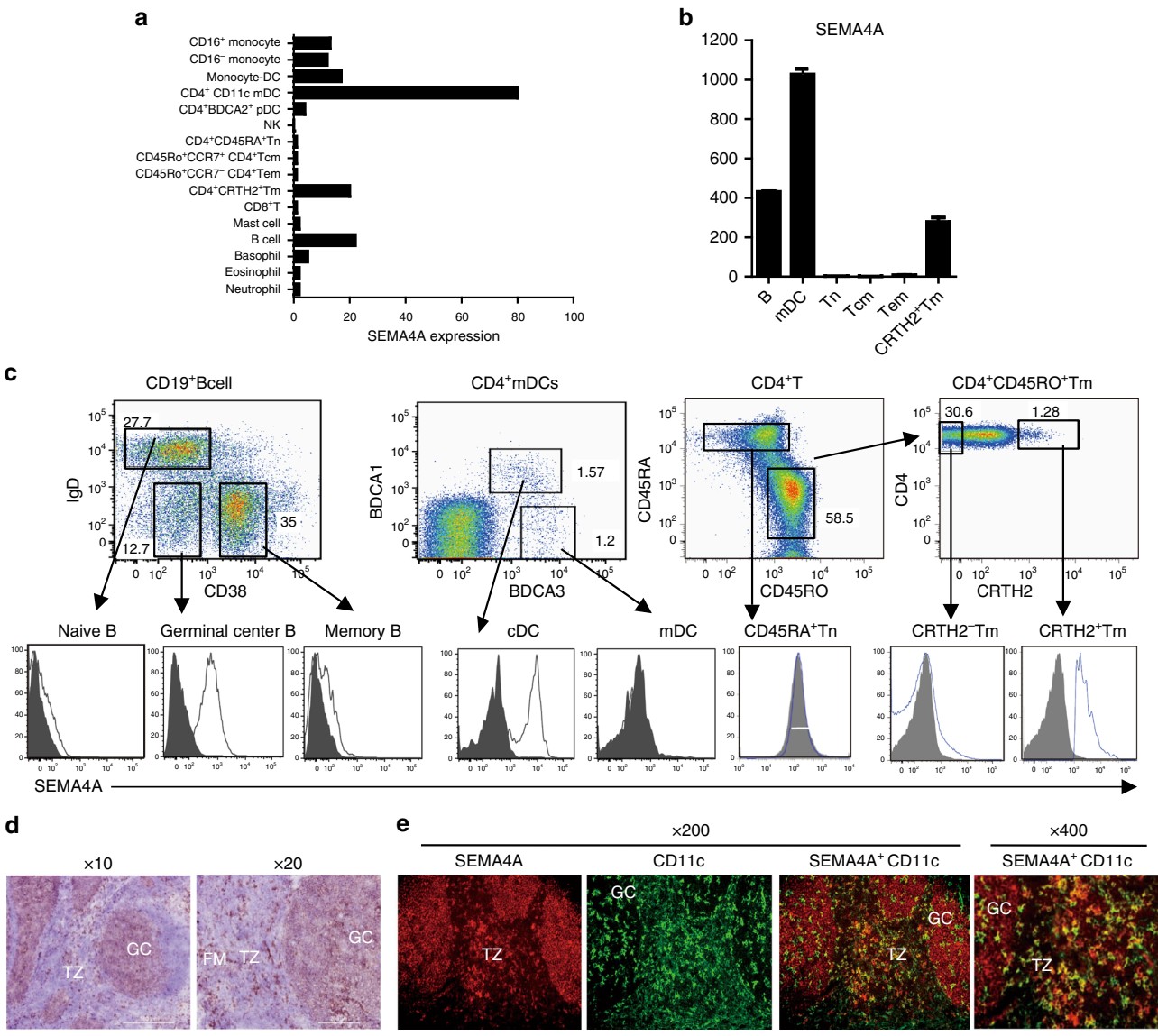

**Fig. 1** SEMA4A is highly expressed in antigen-presenting cells (APCs) and CRTH2[+] Th2 memory T cells. **a**, **b** Quantification of *SEMA4A* mRNA in different human haematopoietic cell types. The indicated human haematopoietic cell types were purified, and total RNA was isolated from each cell population for chip hybridization. Bars show the relative *SEMA4A* mRNA expression in each cell subset (**a**). Real-time PCR was performed to confirm the expression of *SEMA4A* mRNA in CD19[+] B cells, CD4[+]CD11c[+]myeloid dendritic cells (mDCs), as well as naive (Tn), central memory (Tcm), effector memory (Tem), and Th2 memory CD4[+] T cells. The relative fold differences in *SEMA4A* mRNA expression between samples are indicated on the y-axis. Error bars represent the standard error of mean (s.e.m.) of different wells. Data represent one of five donors in each independent experiment, which were repeated three times. **b**, **c** Flow cytometry analysis of SEMA4A expression on different populations of CD19[+] B cells, CD4[+] DC, and CD4[+] T cells. Enriched CD4[+] T cells from peripheral blood are sorted into CD45RA[+] naive, CRTH2[+] memory Th2, and CRTH2[−] memory CD4[+] T cells. CD4[+] conventional dendritic cells (cDCs) are gated as Lin[−]CD4[+]BDCA1[+] cells in CD4[+] enriched tonsil cell; B cells are gated as CD19[+] tonsil cells. The numbers in the dot plots indicate the percentage of cells in that gate. Open histograms represent the staining of indicated cell subsets with anti-SEMA4A mAb; filled histograms represent the isotype. Data represent one of three independent experiments. **d**, **e** Histological analysis of SEMA4A expression in human tonsil. Tonsil tissue sections were stained with anti-SEMA4A mAb followed by Horseradish Peroxidase (HRP)-conjugated goat anti-mouse antibody. Magnifications used for pictures are ×10 (left) with scale bar representing 500 μm and ×20 (right) with scale bar representing 200 μm (**b**); or double immunofluorescence stained with anti-SEMA4A mAb (red) and anti-CD11c mAb (green). Magnifications used for pictures are ×200 (left) with scale bar representing 20 μm and ×400 (right) with scale bar representing 10 μm (**e**). Germinal center (GC), follicular mantle zone (FM), and interfollicular T-cell zone (TZ) were indicated. Data represent one of four independent experiments

or parental L cells in the presence of immobilized anti-CD3, anti-CD28 mAb, or in the presence of immobilized anti-CD3, anti-CD28, anti-IL-4 mAb and IL-12 (Th1-skewing condition), or in the presence of immobilized anti-CD3, anti-CD28, anti-IFN-γ mAb, and IL-4 (Th2-skewing condition) for 7 days[26,27]. Then the activated CD4[+] T cells, or the Th1- or Th2-conditioned CD4[+]

T cells were restimulated with anti-CD3 and anti-CD28 mAbs for 24 h, and the secreted cytokines in the culture supernatants were measured by ELISA. Surprisingly, in contrast to murine Sema4A which drives Th1 response, hSEMA4A significantly promoted production of the cytokines IL-4, IL-5, and IL-13 in activated human CD4[+] T cells ($p < 0.05$, $p < 0.01$, and $p < 0.01$, by Student's

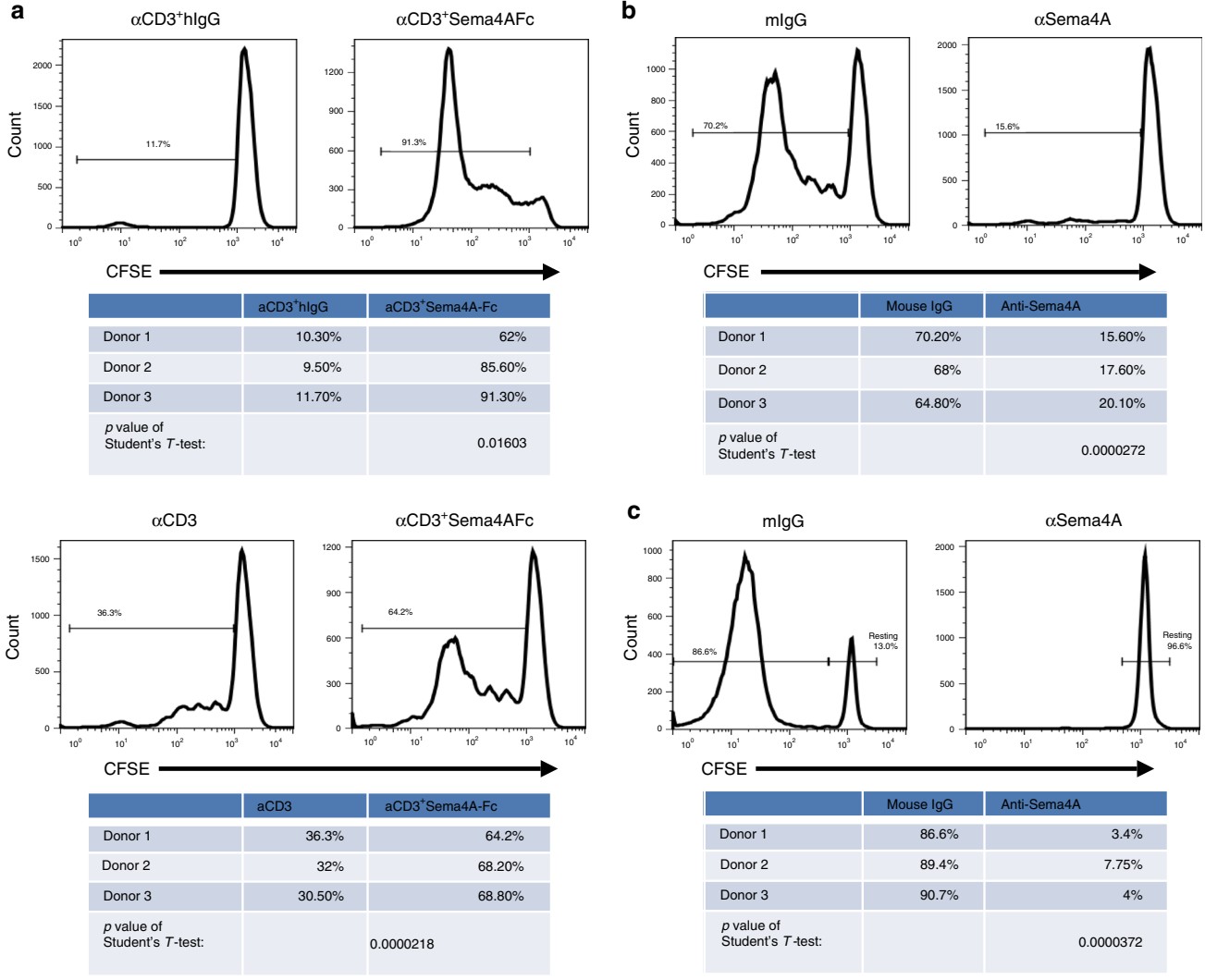

**Fig. 2** SEMA4A co-stimulates T-cell proliferation. **a**, **b** CFSE-labeled purified naive CD4$^+$ T cells were cultured with immobilized suboptimal dose of anti-CD3 mAb, recombinant SEMA4A-Fc fusion protein or hIgG (Fig. 2a upper left), aggregate data were presented at the bottom. Difference between these two groups was evaluated by Student's *T*-test ($p < 0.01603$) or cultured on parental L cells or SEMA4A-expressing L cells pre-coated with a suboptimal dose of anti-CD3 mAb (Fig. 2a down left), aggregate data were presented at the bottom. Difference between these two groups was evaluated by Student's *T*-test ($p < 0.0000218$). FACS data represent one of three independent experiments, respectively. Meanwhile, CFSE-labeled purified naive CD4$^+$ T cells were cultured with immobilized suboptimal dose of anti-CD3 mAb in the presence of anti-SEMA4A mAb or mIgG (**b**) for 5 days. T-cell proliferation was detected by CFSE dilution. Data represent one of three independent experiments. Aggregate data were presented at the bottom. Student's *T*-test was used for the statistical analysis, $p < 0.0000272$. **c** CFSE-labeled purified naive CD4$^+$ T cells were cultured with allogeneic CD4$^+$mDC at a T:mDCs cell ratio of 5:1 in the presence of anti-SEMA4A mAb or mIgG for 7 days. T-cell proliferation was detected by CFSE dilution. Data represent one of three independent experiments. Aggregate data were presented at the bottom. Student's *T*-test was used for the statistical analysis, $p < 0.0000372$

*T*-test). But IFN-γ production was not significant increased ($p > 0.05$, by Student's *T*-test) (Fig. 3a). Most strikingly, SEMA4A simulated robust production of IL-4, IL-5, and IL-13 in the Th2-skewing condition ($p < 0.05$, $p < 0.0001$, and $p < 0.0001$, by Student's *T*-test), and inhibited IFN-γ production in the Th1-skewing condition ($p < 0.001$, by Student's *T*-test) (Fig. 3b). The same results were obtained from the purified CRTH2$^+$CD4$^+$Th2 memory cells cultured in the same conditions (IL-4, $p < 0.001$; IL-5, $p < 0.0001$; IL-13, and $p < 0.001$; IFN-γ, $p < 0.0001$; by Student's *T*-test) (Supplementary Fig. 2a). The production of Th2 cytokines promoted by SEMA4A-espressing L cells in the Th2-skewing condition could be completely blocked by anti-SEMA4A mAb in naive CD4$^+$T cells (IL-4, $p < 0.001$; IL-5, $p < 0.01$; IL-13, $p < 0.01$; by Student's *T*-test) and in CRTH2$^+$Tm cells (IL-4, $p < 0.05$; IL-5, $p < 0.0001$; IL-13, $p < 0.05$; by Student's *T*-test) (Fig. 3c and Supplementary Fig. 2b).

Differentiation of T cells into Th1 or Th2 cells crucially depends on the relative expression of the transcriptional regulators T-bet and GATA3[28–30]. Therefore, we detected the expression of T-bet and GATA3 in the cultured CD4$^+$ T cells by q-PCR. T-cell activation-induced upregulation of GATA3 and T-bet. SEMA4A expressed on L cells further increased GATA3 expression in Th2-conditioned cell ($p < 0.05$, by Student's *T*-test). On the contrary, SEMA4A not only inhibited T-bet expression in activated CD4$^+$ T cell, and Th1-conditioned cells compared with the cells cocultured with parental L cells ($p < 0.05$, by Student's *T*-test), but also more severely inhibited the T-bet expression in the Th2-conditioned cells although T-bet has been markedly suppressed in Th2-conditioned cells cocultured with parental L cells ($p < 0.05$, by Student's *T*-test) (Fig. 3d). In addition, SEMA4A strongly promoted STAT6 phosphorylation (Fig. 3e, Supplementay Fig. 6), which is required for IL-4-induced Th2

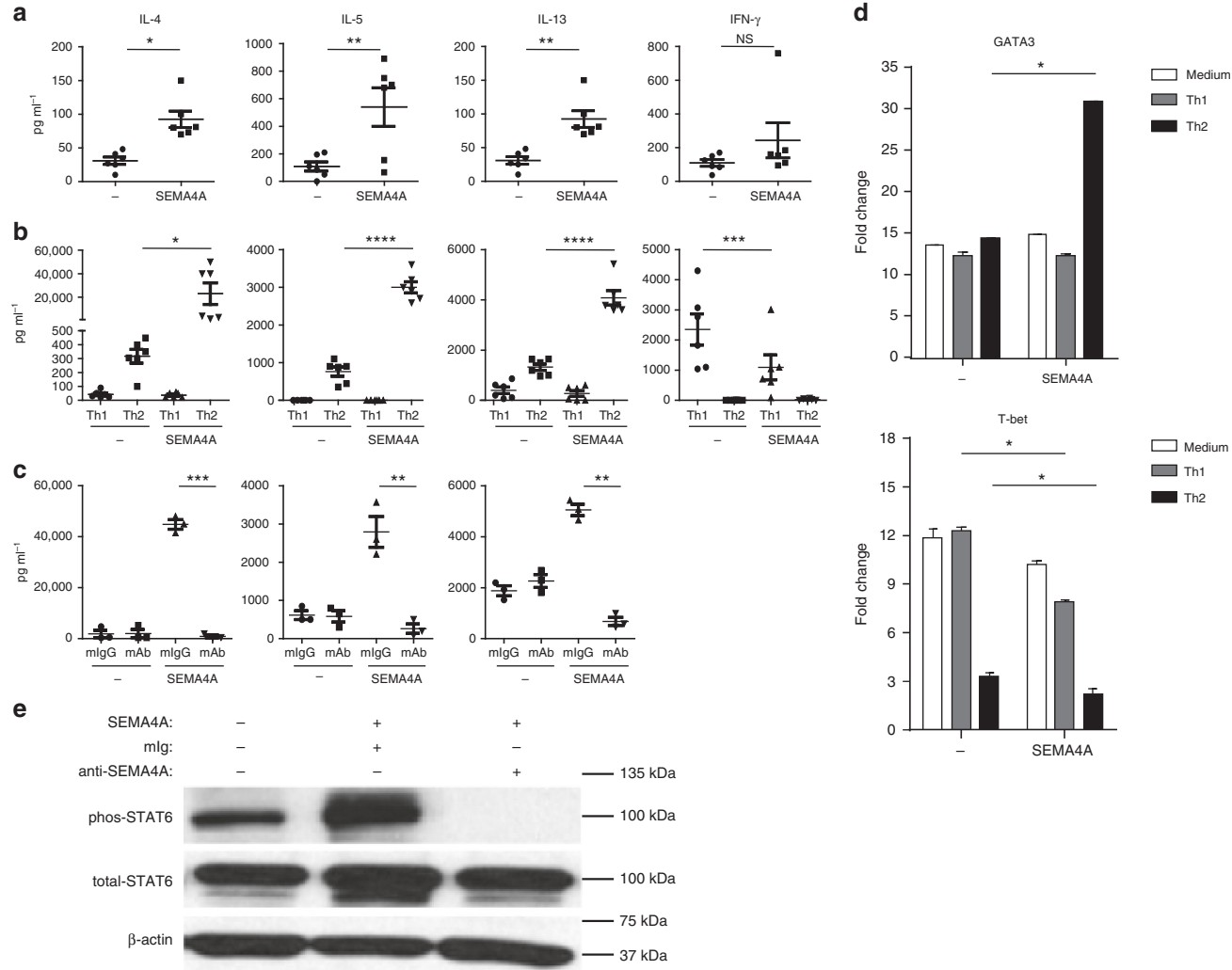

**Fig. 3** SEMA4A enhances Th2 polarization of Th2-primed CD4$^+$ T cells. **a–c** SEMA4A enhances Th2 cytokine production. The purified naive CD4$^+$ T cells were cultured on parental L cells or SEMA4A-expressing L cells pre-coated with a suboptimal dose of anti-CD3 mAb, anti-CD28 mAb (**a**), or in Th1- or Th2-skewing conditions (**b**), or in Th2-skewing condition with anti-SEMA4A mAb or mIgG (**c**) for 7 days. Cultured T cells were collected and restimulated with immobilized anti-CD3, anti-CD28 for 24 h before the measurements of cytokines in the supernatants by ELISA. Each dot represents one donor. Data represent one of five independent experiments. Student's *T*-tests were used for statistical analysis, *p*-value was as shown, *$p < 0.05$, **$p < 0.01$, ***$p < 0.001$, and ****$p < 0.0001$. **d** SEMA4A upregulates expression of GATA3 and downregulates expression of T-bet. Purified naive CD4$^+$ T cells were cultured as in **b** for 2 days, and the gene expression levels were measured by real-time PCR. The relative fold differences in gene expression between samples are indicated on the *y*-axis. Error bars represent the s.e.m. of different wells. Data represent one of five independent experiments. **e** SEMA4A enhances phosphorylation of SATA6. Purified naive CD4$^+$ T cells were cultured as in **c** for 7 days, and then restimulated with anti-CD3, anti-CD28 for 30 min. Activated STAT6 was detected by western blot with β-actin serving as an internal control. Data represent one of four independent experiments

differentiation[31,32]. These results demonstrated that, in contrast to mouse Sema4A, hSEMA4A promote Th2 responses instead of Th1 responses.

**ILT-4 is the receptor on activated human T cells for SEMA4A.** To determine the cellular expression of human SEAM4A receptors, we stained human CD4$^+$ T cells with SEMA4A-Fc protein. SEMA4A-Fc bound the freshly isolated human peripheral blood CD45RO$^+$CRTH2$^+$CD4$^+$memory Th2 cells, but not naive T cells or CRTH2$^-$ memory CD4$^+$T cells (Fig. 4a). After activation with anti-CD3, anti-CD28, the three populations of CD4$^+$ T cells were bound by SEMA4A-Fc with the highest binding of SEMA4A-Fc on activated memory Th2 cells (Fig. 4a). These data demonstrated that the receptor of SEMA4A is expressed on activated CD4$^+$ T cell, and preferentially expressed on Th2 cells. Next, we adopted

two approaches to clone the putative hSEMA4A receptor on activated CD4$^+$ T cells. One way is to construct an cDNA expression library from anti-CD3/anti-CD28-activated-CD4$^+$ T cells in a retroviral system[33]. Human CD4$^+$ lymphoma H9 cells were transfected with the cDNA library and stained with an Allophycocyanin (APC)-conjugated SEMA4A-Fc protein. The SEMA4A-Fc-binding cells were enriched through several rounds of cell sorting (Fig. 4b). Genomic DNA was isolated from the enriched SEMA4A-Fc-binding cells followed by PCR cloning. A discrete band corresponding to an insert of roughly 2.2 kb was seen (Fig. 4c). Sequencing of the cDNA insert identified the full-length cDNA encoding ILT-4 receptor. Another way is to employ a receptor assay[34], in which SEMA4A-Fc was used as a fishing bait to screen a cDNA library encoding more than 2000 full-length human transmembrane molecules expressed on immune or haematopoietic cells. The binding of SEMA4A to ILT-4-

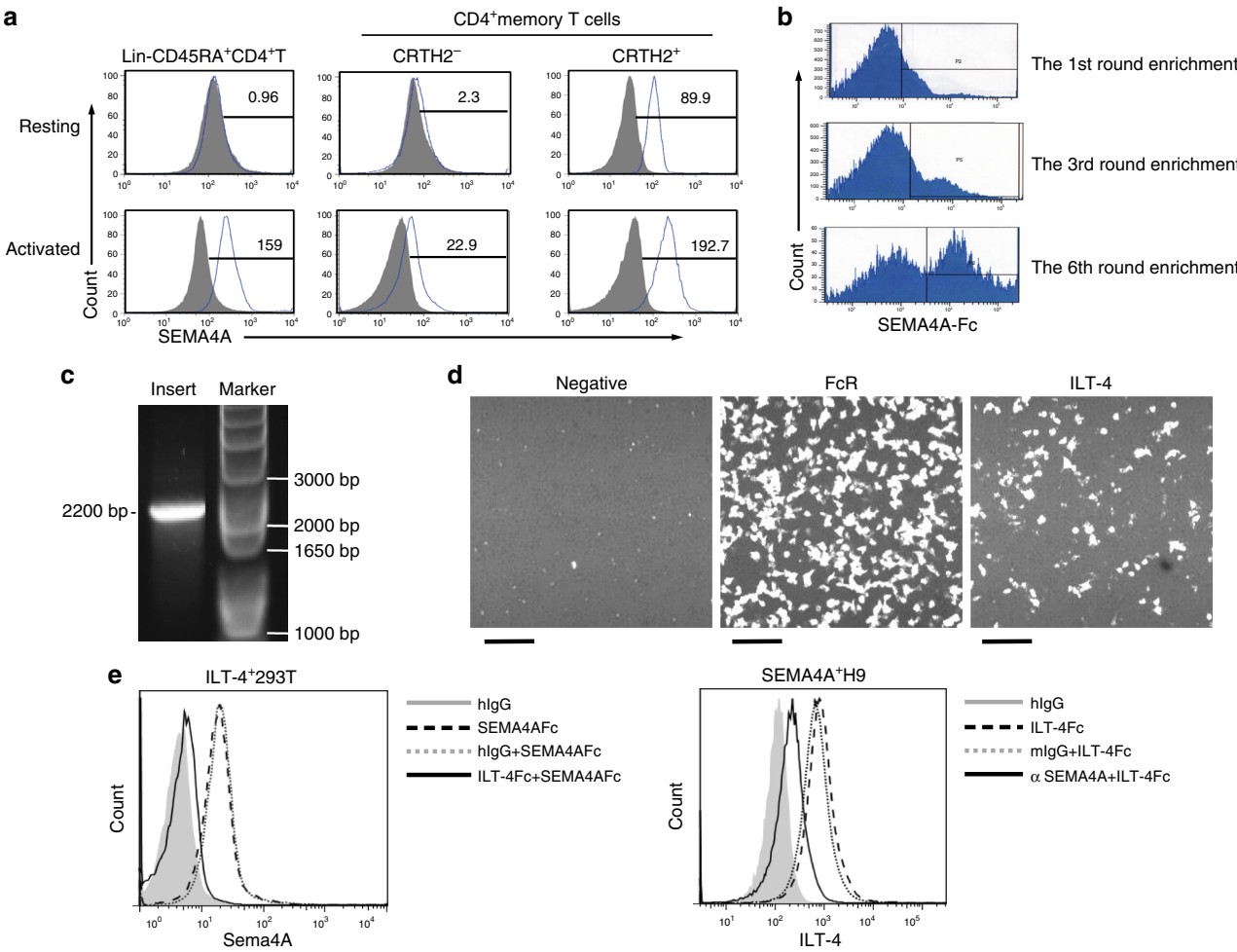

**Fig. 4** ILT-4 is identified as a receptor for human SEMA4A. **a** Binding of SEMA4A-Fc to activated T cells. Freshly isolated naive (lin⁻CD45RA⁺), memory Th2 (CD45RO⁺CRTH2⁺) and CD45RO⁺CRTH2⁻CD4⁺ T cells were stained with SEMA4A-Fc (upper), or activated with anti-CD3 mAb⁺ anti-CD28 mAb for 1 day before staining with SEMA4A-Fc (down). SEMA4A-Fc binding was analyzed by flow cytometry. Open histograms represent the staining of indicated cell subsets with SEMA4A-Fc; filled histograms represent the isotype. Numbers in the histograms indicate the mean fluorescence intensity for the markers indicated below histograms. Data represent one of five independent experiments. **b** Expression cloning of SEMA4A receptor. H9 cells were infected with retroviral cDNA library of activated CD4⁺ T cells, and the SEMA4A-Fc-binding cells were enriched by flow cytometry. **c** Flow cytometry enriches a cDNA insert of about 2.5 kb. The genome for the enriched cells were isolated and the inserts were amplified by PCR with a pair of primers designed according to the sequence of the retroviral expression vector. **d** Identification of interaction between ILT-4 and SEMA4A by high-throughput screening of a receptor array. 293T cells were transfected with plasmids of different human transmembrane genes in different wells of 384-well plates. SEMA4A-Fc was added into each well to evaluate its binding to the transfects by Applied Biosystems 8200 Cellular Detection System. The Fc receptor transfectants are used as positive control. Scale bars represent 50 μm. **e** Interaction of SEMA4A and hILT-4 was tested by FACS. 293T cells were transfected with ILT-4 cDNA and stained with hIgG (filled histogram), SEMA4A-Fc (dash line), SEMA4A-Fc+ an excess of ILT-4-Fc (full line), or SEMA4A-Fc+ an excess of mIgG (dotted line) (left); H9 cells were transfected with *SEMA4A* cDNA and stained with hIgG (filled histogram), ILT-4-Fc (dash line), an excess of anti-SEMA4A mAb + ILT-4-Fc (thick line), or an excess of mIgG+ SEMA4A-Fc (dotted line) (right). Data represent one of five independent experiments

transfected cells was revealed in this assay (Fig. 4d). The interaction of SEMA4A and ILT-4 was further confirmed by binding of the SEMA4A-Fc to ILT-4-transfected 293T cells, which could be blocked by ILT-4-Fc (Fig. 4e, left). Conversely, ILT-4-Fc-bound to Sema4A-transfected H9 cells, which could be blocked by anti-SEMA4A mAb (Fig. 4e, right). Using surface plasmon resonance, we estimated the dissociation constant (Kd) of SEMA4A and ILT-4 to be 3.8 μM (Supplementary Fig. 3). These data demonstrated that ILT-4 is a receptor expressed on activated CD4⁺ T cells for SEMA4A.

ILT-4 belongs to a family of ILT, and has been shown to be predominantly expressed on myeloid lineage cells. Its expression on T cells has not been reported. To test whether ILT-4 is the

receptor for SEMA4A on CD4⁺ T cells, we first checked the expression of mRNA of ILT-4 in purified CD4⁺ T cells by RT-PCR. ILT-4 mRNA was rarely detected in naive CD4⁺ T cells. However, the expression of ILT-4 mRNA was dramatically upregulated in T-cells activated with anti-CD3 and anti-CD28, and the expression intensity continued to increase in 4-day culture (Fig. 5a). ILT-4 was not expressed on naive CD4⁺ T cells, but inducibly expressed on activated CD4⁺ T cells with anti-CD3 and anti-CD28 antibodies (Fig. 5d). Moreover, SEMA4A-Fc could block the binding of anti-ILT-4 antibody to activated T cells (Fig. 5b). Consistent with the expression of ILT-4 on CD4⁺ T cells, SEMA4A-Fc did not bind to naive CD4⁺ T cells, but strongly bound to activated CD4⁺ T cells. Moreover, the binding

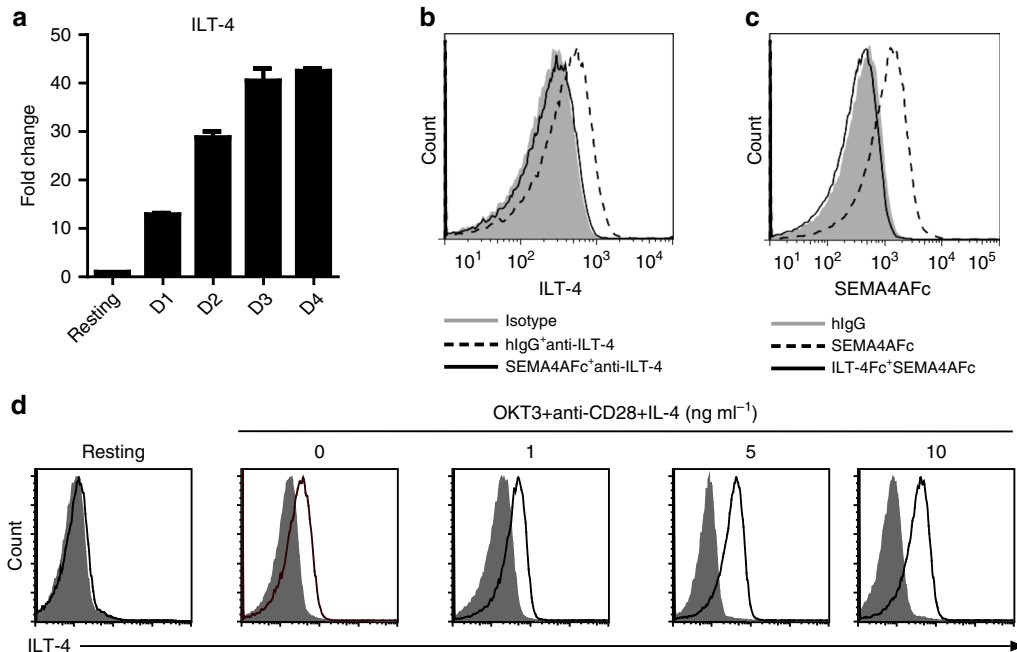

**Fig. 5** ILT-4 is a receptor inducibly expressed on activated human CD4$^+$ T cells for SEMA4A. **a** Expression of ILT-4 mRNA was induced in activated CD4$^+$ T cells. The expression levels of ILT-4 mRNA were detected by real-time PCR in freshly purified human naive CD4$^+$T cells or Tn cells activated with immobilized OKT3 and soluble anti-CD28 mAb for the indicated day1, day 2, day 3, and day 4, respectively. The relative fold differences in gene expression between samples are indicated on the y-axis. Error bars represent the s.e.m. of different wells. Data represent one of four independent experiments. **b** SEMA4A-Fc blocks the binding of anti-ILT-4 mAb to activated CD4$^+$ T cells. Purified naive CD4$^+$ T cells were activated with immobilized anti-CD3, anti-CD28 mAb for 1 day. Cells were stained with isotype control antibody (filled area), anti-ILT-4 mAb plus hIgG (dashed line), or an excess of SEMA4A-Fc plus anti-ILT-4 mAb (full line), and analyzed by flow cytometry. Data represent one of five independent experiments. **c** ILT-4-Fc blocks the binding of SEMA4A-Fc to activated CD4$^+$ T cells. Purified naive CD4$^+$ T cells were activated with immobilized anti-CD3, anti-CD28 mAbs for 1 day. Cells were stained with hIgG (filled histogram), SEMA4A-Fc (dashed line) or an excess of ILT-4-Fc and SEMA4A-Fc (full line), and analyzed by flow cytometry. Data represent one of five independent experiments. **d** IL-4 enhances ILT-4 expression on activated T cells. Purified naive CD4$^+$ T cells were activated with immobilized anti-CD3 plus anti-CD28 mAbs in the absence or presence of IL-4 at the indicated concentrations for 7 days. ILT-4 expression was detected by flow cytometry. Open histograms represent the staining with anti-ILT-4 mAb; filled histograms represent the isotype. Data represent one of five independent experiments

of SEMA4A to activated CD4$^+$ T cells could be blocked by ILT-4-Fc (Fig. 5c). These data demonstrated that ILT-4 is a receptor expressed on activated CD4$^+$ T cells for Sema4A.

To directly test whether ILT-4 mediate the effect of SEMA4A on CD4$^+$ T-cell response, CSFE-labeled naive CD4$^+$ T cells were cultured with immobilized anti-CD3 mAb plus SEMA4A-Fc in the presence or absence of soluble ILT-4-Fc. T-cell proliferation was detected with a CFSE dilution assay after 7 days of culture. ILT-4-Fc blocked CD4$^+$ T-cell proliferation co-stimulated by SEMA4A in a dose-dependent manner ($p < 0.01$, by Student's T-test) (Fig. 6a). Naive CD4$^+$ T cells were cocultured with SEMA4A-expressing L cells coated with suboptimal dose of anti-CD3 mAb in the presence or absence of anti-ILT-4 antibody. Similarly, anti-ILT-4 blocked SEMA4A-mediated CD4$^+$ T-cell proliferation in a dose-dependent manner ($p < 0.05$, by Student's T-test) (Fig. 6b). These results demonstrated that SEMA4A co-stimulates CD4$^+$ T-cell responses through receptor ILT-4. Due to the preferentially binding of SEMA4A-Fc to CD45RO$^+$CRTH2$^+$ CD4$^+$memory Th2 cells (Fig. 4a), we wonder whether Th2 cytokine can induce the expression of ILT-4. We found that IL-4 treatment increased ILT-4 expression on activated T cells in a dose-dependent manner (Fig. 5d). To further define the potential roles of SEMA4A in the pathophysiology of human allergic disease, we examined mRNA and protein expression of Sema4A in lung tissues obtained from patients with allergic asthma and normal lung tissues. Both Q-PCR ($p < 0.001$, by Student's T-test) and immunohistochemical staining showed that SEMA4A

expression is significantly higher in asthmatic lung compared to healthy lung. SEMA4A was expressed in clusters of infiltrating cells in asthmatic lung tissue (Supplementary Fig. 4). Meanwhile, within asthmatic lung tissue infiltrated cells, CD4$^+$ T cells expressed ILT-4 (Supplementary Fig. 5).

## Discussion

Sema4A belongs to the large semaphorin family that is a new class of immune regulatory molecules. In murine immune system, Sema4A is constitutively expressed on APCs, and co-stimulates proliferation and cytokine production of T cells. In sema4A-deficient mouse, T cells have selective defect in the Th1 differentiation in vitro and in vivo[8,15]. However, the functions of SEMA4A in human immune system is unknown. In this study, we investigated the expression of SEMA4A in human immune cells and its roles in regulation of human CD4$^+$ T-cell responses.

By using microarray, Q-PCR, flow cytometry and immuno-histochemistry, we identified the high expression of SEMA4A on human mDC, B cells and activated T cells, but not on naive T cells, which is consistent with the expression of murine Sema4A on immune cells. Similar to murine Sema4A, hSEMA4A co-stimulates T-cell proliferation. However, their roles in differentiation of activated CD4$^+$ T cell might be different. Soluble murine Sema4A promotes the induction of either Th1-cell-mediated IFN-γ production or Th2-cell-mediated IL-4 production following stimulation with anti-CD3 mAb, depending

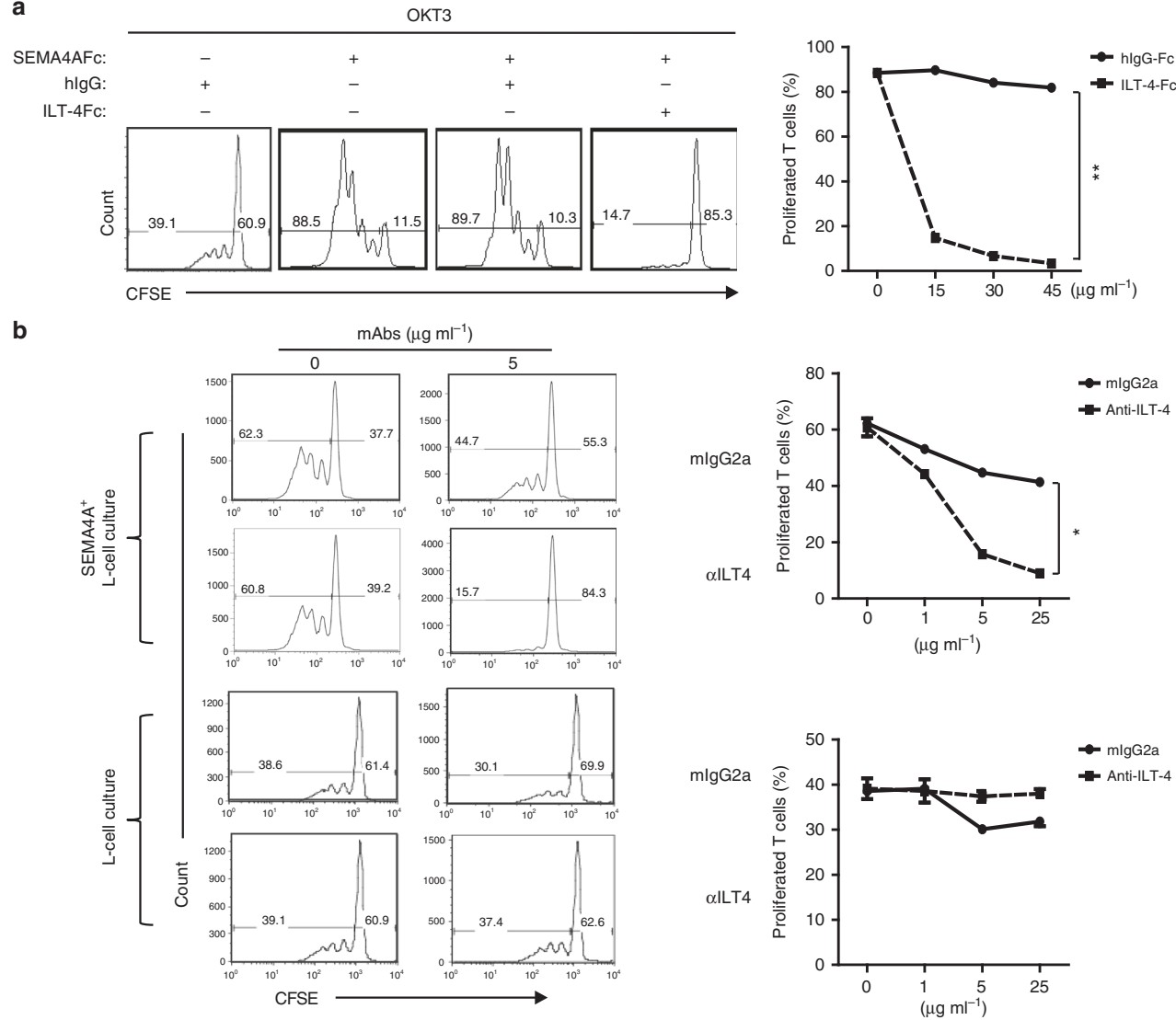

**Fig. 6** SEMA4A co-stimulates CD4$^+$ T-cell proliferation through ILT-4 receptor. **a**, **b** CFSE-labeled purified naive CD4$^+$T cells were cultured in the presence of suboptimal dose of immobilized anti-CD3 mAb plus human SEMA4A-Fc or control hIgG with or without various concentrations of ILT-4-Fc or hIgG for 5 days **a**, or cultured on parental L cells or SEMA4A-expressing L cells pre-coated with a suboptimal dose of anti-CD3 mAb in the presence of various concentrations of anti-ILT-4 or mIgG for 7 days **b**. T-cell proliferation was detected by FACS analysis; the percentages of proliferated and non-proliferated T cells are indicated in each squares. The percentages of proliferated T cells in the indicated group were summarized as proliferation blocking curve ($p <$ 0.01) (Student's *T*-tests). Data represent one of three independent experiments

on the respective culture conditions[7]. Sema4A-deficient murine T cells fail to differentiate into IFN-γ-producing cell in Th1-skewing conditions, and normally differentiate into IL-4-producing cell in Th2-skewiing conditions. The generation of IFN-γ-producing antigen-specific T cells is impaired in Sema4A-deficient mice immunized with Th1-inducing agents, such as heat-killed *Propionibacterium acnes*. Conversely, Sema4A-di.ficient mice mount enhanced Th2 responses when infected with a Th2-inducing intestinal nematode, *Nippostrongylus brasiliensis*[8]. Our study demonstrated that hSEMA4A coated on the surface of plates or L cells remarkably promotes production of Th2 cytokines IL-4, IL-5, and IL-13, but not Th1 cytokine IFN-γ in activated human CD4$^+$ T cells. Furthermore, SEMA4A simulated robust production of IL-4, IL-5, and IL-13 in the Th2-skewing condition, and inhibited IFN-γ production in the Th1-skewing condition. The effect of SEMA4A on Th2 differentiation is supported by the upregulation of GATA3 expression

and STAT6 phosphorylation, and inhibition of T-bet expression induced by SEMA4A.

Sema4A binds to plexinBs, plexin D1, Nrp1, and Tim-2, and each of these receptors mediates distinct functions. In mouse model, Sema4A is required for the function and stability of regulatory T (Treg) cells by binding to Treg-cell-expressed receptor Nrp1[21,22]. However, in human, higher NRP1 expression was observed on CD4$^+$ T cells isolated from secondary lymphoid tissue compared with peripheral blood[35–37]. NRP$^+$ Treg was found in synovial fluid of rheumatoid arthritis patients[38] and inflamed muscles[39]. In some cancer patient, NRP1 was upregulated in Treg[35,37,40]. Tim-2 expression is highly restricted to activated T cells. Because Sema4A binding induces tyrosine phosphorylation of the cytoplasmic tail of Tim-2, Tim-2 seems to transduce Sema4A signals[7]. The similar phenotypes of Tim-2- and Sema4A-deficient mice support the idea that Tim-2 serves as a functional receptor on activated T cells for Sema4A[41].

But, there are also some inconsistencies. For example, T cells from Tim-2-deficient mice but not those from Sema4A-deficient mice show enhanced basal proliferation. Thus, it is likely that Sema4A or Tim-2 has another binding partner in activated T cells. Actually, the functions of Tim-2 by binding to Sema4A has not been directly tested. In this study, we applied two different techniques independently performed in two laboratories to clone the receptor for SEMA4A from human T cells, both of which identified ILT-4 as a receptor for hSEMA4A.

ILT-4 belongs to a family of Ig-like transcripts (ILTs, also referred to as LILR, LIR, and CD85), which represent an Ig type of activating and inhibitory receptors coded by more than 10 genes located in the 19q13.4 chromosome[42]. ILT-4 is composed of four extracellular Ig-like domains and three immune-receptor tyrosine-based inhibitory motifs (ITIMs) in cytoplasmic part. It is largely expressed mainly by the myelomonocytic lineage, such as monocytes, macrophages, and DCs[43–45]. ILT-4 interacts with several HLA class I molecules but binds the non-classical HLA class I molecule HLA-G with the highest affinity[46]. Interaction of ILT-4 with HLA-G had been shown to inhibit the differentiation of monocytes into dentritic cells, and induce the development of tolerogenic DC with the consequent induction of immunosuppressive T cells[47,48]. ILT-4 can bind to soluble complement fragment C4d to act as a cellular receptor. Interaction between cell surface-resident ILT-4 and soluble monomeric C4d resulted in endocytosis of C4d[49]. The expression of ILT-4 on T cells has not been reported. Our study showed that ILT-4 was not expressed on naive CD4[+] T cells, but inducibly expressed on activated CD4[+] T cells. The binding of anti-ILT-4 mAb or SEMA4A-Fc to activated T cells could be blocked by SEMA4A-Fc or anti-ILT-4 mAb respectively, suggesting that ILT-4 might be the only receptor for SEMA4A on activated CD4[+] T cells. Furthermore, IL-4 could enhance ILT-4 expression on activated T cells in a dose-dependent manner. This is consistent with the promotion of Th2 differentiation by SEMA4A co-stimulation.

In summary, we have shown that hSEMA4A, which is mainly expressed on APCs, enhances activation and Th2 differentiation of CD4[+] T cells and identified ILT-4 as a receptor for hSEMA4A on activated CD4[+] T cells.

## Methods

**Cell line culture**. Human HEK293T cells (ATCC CRL-11268) were cultured in Dulbecco's modified Eagle's medium. Mouse L cells (ATCC CRL-2648), Human H9 cells (ATCC HTB-176), purified human CD4[+]CD45RA[+]Tn cells, and CD4[+]CD11c[+]mDCs were cultured in RPMI-1640 medium[+] GlutaMAX-I medium (Gibco). All growth media were supplemented with 10% (vol vol[−1]) FBS (Sigma or Atlanta), penicillin (100 U ml[−1]), and streptomycin (100 μg ml[−1]). All cell types were maintained in a humidified atmosphere at 37 °C in a 5% $CO_2$ incubator.

**Purification of T B DC cells and other lineages**. Research protocol of using adult blood buffy coats of this study were approved by the institutional review board at The University of Texas, M.D. Anderson Cancer Center (Houston, TX) (LAB-03-0390). Unidentified adults blood buffy coats from healthy donors were only encoded by numbers without any personal informations were obtained from the Gulf Coast Regional Blood Center, Houston, TX. CD4[+] T cells were enriched from blood buffy coats by means of the CD4[+] T-cell isolation kit (Stem cell, 15022 or 15062) according to manufacturer's instructions. Enriched CD4[+] T cells were stained with FITC-labeled lineage cocktail of antibody CD8 (HIT8a, dilution 1:100), CD14 (MφP9, dilution 1:100), CD16 (3G8, dilution 1:100), CD19 (HIB19, dilution 1:100), CD20 (2H7, dilution 1:100), CD56 (B159, dilution 1:100), CD11c (B-Ly6, dilution 1:100), TCRγ/δ (11F2,dilution 1:100) from BD Biosciences; CD25 (VT-072, dilution 1:100) from Biolegend; PE-CD45RO (UCHL1,dilution 1:200), APC-cy7-CD4 (RPA-T4, dilution 1:200), PerCP-Cy5.5-CD45RA (HI100, dilution 1:100), CD11c-APC (B-ly6, dilution 1:100) from BD Biosciences; Biotin-CRTH2 (BM16, 1:200 dilution) from MilteneyBiotec, and then washed, revealed with strepavidin-PE-cy7 from ebiosciences. Stained cells were sorted into fractions of CD45RA[+]CD4[+] naive T, CD45RO[+]CRTH2[+]CD4[+] Th2 memory, CD45RO[+]CRTH2[−]CD4[+] Th1 memory, and CD4[+]CD11c[+]DCs by using a FACSAria (BD Biosciences) with purity >99%. pDC was enriched firstly by adding lineage cocktail mAbs of anti-CD3 (SP34-2, dilution 1:100), CD14 (MφP9, dilution 1:100), CD11b (M1/70, dilution 1:100), CD19 (HIB19, dilution 1:100), CD57 (NK-1, dilution 1:100), CD235a (GA-R2 (HIR2), dilution 1:100) from BD Biosciences, following by adding anti-mIgG MicroBeads (MilteneyBiotec,130-048-401) and going through LD columns (MilteneyBiotec, 30-042-901). Enriched pDC was stained with antibodies of APC-cy7-CD4 (RPA-T4, dilution 1:200), BV421-BDCA2 (V24-785, dilution 1:200), CD11c-APC (B-ly6, dilution 1;100), and FITC-lineage anti-(CD3, CD14, CD11b, CD19, CD57, CD235a) from BD Biosciences and were sorted into fraction of CD4[+]BDCA2[+]pDC. Eosinophils were enriched by two layer percoll gradient centrifugation of buffy coats blood first, then purified by adding CD15 microbeads (MilteneyBiotec, 130-046-601) following run through LD columns. CD15-depleted cells were collected and centrifuged. Basophils were isolated by using EasySep Human Basophil Enrichment Kit (Stem cells,19069) according to manufacturer's instructions. Mast cells were generated from human CD34[+] stem cells. The following peripheral blood cell subsets were freshly sorted as: CD16[+] monocytes and CD16[+] monocyte-derived DCs (monocytes cultured for 6 days with GM-CSF and IL-4), CD68[+]CD16[+] neutrophils, CD16[+]CD56[+] NK cells, CD19[+] B cells, and CD3[+]CD8[+] T cells.

**T-cell activation and proliferation**. For T-cell activation, purified CD4[+]CD45RA[+] naive T cells were cultured with immobilized anti-CD3 (OKT3, 2 μg ml[−1]) and soluble anti-CD28 (L293.1, 1 μg ml[−1]) in the absence or presence of IL-4 (10 ng ml[−1]) in 96-well culture plates. For T-cell co-stimulation, purified CD4[+] naive T cells were labeled with CFSE (Life Technologies), and cultured either in the presence of immobilized anit-CD3 at 2.0 μg ml[−1] and recombinant SEMA4A-Fc protein (R&D systems) at 5.0 μg ml[−1], or on irradiated CD32-expressing parental L cells or CD32/SEMA4A-expressing L cells (SEMA4A-L cells) pre-coated with anti-CD3 mAb (OKT3; 0.2 μg ml[−1]) in the absence or presence of recombinant ILT-4-Fc protein (R&D system), or anti-hSEMA4A (house hold) or anti-ILT-4 (287219, R&D system) antibodies for 5 days. T-cell expansion was analyzed by FACS according to CFSE dilution and cell number counts.

For MLRs, CD4[+] naive T cells were labeled with CFSE and cocultured with CD4[+]CD11c[+] mDCs at an mDC to T-cell ratio of 1:5 in the absence or presence of anti-SEMA4A antibody or mIgG for 7 days. T-cell proliferation was detected by CFSE dilution through FACS analysis.

**Cytokine production analysis**. Purified CD4[+] naive T cells were cultured on irradiated CD32-expressing parental L cells or CD32/SEMA4A-expressing L cells (SEMA4A-L cells) pre-coated with anti-CD3 antibody (OKT3; 0.2 μg ml[−1]) and anti-CD28 (1 μg ml[−1]) in the absence or presence of anti-IL-4 (5 μg ml[−1]) plus IL-2 (10 ng ml[−1]) (Th1-skewing conditions) or anti-IFN-γ (5 μg ml[−1]) plus IL-4 (10 ng ml[−1]) (Th2-skewing condition) for 7 days. Cultured T cells were collected and then restimulated with immobilized anti-CD3 (2 μg ml[−1]) and anti-CD28 (1.0 μg ml[−1]) for 24 h. The levels of IL-4, IL-5, IL-13, and IFN-γ in the supernatants were measured by ELISA (R&D Systems) according to the manufacturer's instructions.

**Microarray analysis**. The total RNA was isolated from the purified human cell subset with an RNeasy kit (QIAGEN) according to the manufacturer's instructions and then sent to the Biosense company for chip hybrid analysis. Biotinylated cRNA samples were generated with the Bioarray High-Yield RNA Transcript Labeling kit (ENZO Diagnostics), and purified with RNeasy mini columns (Qiagen) according to the manufacture's protocol. Following fragmentation, 15 μg cRNA was hybridized for 16 h at 45 °C on Affymetrix GeneChip U133[+] 2.0 arrays. Arrays were stained with a streptavidin-phycoerythrin conjugate (molecular probes) and visualized with a GeneArray scanner 3000. The scanned images were aligned and analyzed using the GeneChip software Microarray Suite 5.0 (Affymetrix) according to the manufacturer's instructions. The signal intensities were normalized to the mean intensity of all the genes represented on the array, and global scaling (scaling to all probe sets) was applied before performing comparison analysis. Genes with variable expression levels were selected based on the following criteria: genes should be expressed (have presence calls) in at least one of the three samples and σi/μi ratio should be >0.65, where σi and μi are the standard deviation and mean of the hybridization intensity values of each particular gene across all samples, respectively.

**Real-time quantitative PCR assays**. Total RNA was isolated from purified subsets of PBMCs, stimulated CD4[+] T cells and human normal and asthma lung tissues (from Dr. Ying Sun, England) by RNeasy kit (QIAGEN). The cDNA templates were synthesized using SuperScript II (Life Technologies). TaqMan pre-labeled primers (Life Technologies, Applied Biosystems) were applied, and real-time quantitative PCR (Q-PCR) amplification was performed with the ABI Prism 7500 Life Technologies, Applied Biosystems) detection system. The following primers were used: hSema4A, 5′-ATGCCCAGGGTCAGATACT-3′ and 5′-CATCACCA CTCAGG- AGCAGA-3′; GATA3, 5′-GAAGGCATCCAGACCAGAAA-3′ and 5′-GCTGTTCTTGGGGAAGTC- CT-3′; T-bet, 5′-GAGGCTGAGTTTCGAGC AGT-3′ and 5′-CTGGCCTCGGTAGTAGGACA-3′; GAPDH, 5′-TGCACCACC AACTGCTTAGC-3′ and 5′-GGCATGGACTGTGGTCATGAG-3′; β-actin, 5′-CTGGAACGGTGAAGGTGACA-3′ and 5′-AAGGGACTTCCTGTAACAA TGCA-3′. The relative expression of target genes was normalized using internal controls of β-actin and GAPDH.

**Receptor cloning by cDNA library screening**. Purified human naive CD4+ T cells were activated with immobilized anti-CD3 (2.0 µg ml⁻¹) and anti-CD28 (1.0 µg ml⁻¹) for 24 h. Total RNA was extracted from the activated T cells with an RNeasy kit (QIAGEN) and sent to the ATGC Company for cDNA library construction with retroviral expression vector. The cDNA library were co-transfected with pVSV-G vector into a GP2-293 packaging cell line (Clontech) by Lipofectamine 2000 (Life Technologies, Invitrogen). After 48 h, the replication-incompetent virus particles were collected and condensed. H9 cells were infected with the recombinant retrovirus particles and collected 48–72 h after infection. The cells were stained with an APC-conjugated SEMA4A-Fc fusion protein (Zenon labeling kit, Life Technologies) and applied to FACSAria for enrichment of the SEMA4A-Fc-binding cells. The enriched H9 cells were expanded in vitro culture to sufficient number and sorted again for enrichment of the SEMA4A-Fc-binding cells. After six rounds of sorting when an obvious positive peak of cells were collected, the enriched H9 cells were lysed and the genome DNA was isolated (QIAGEN). The insert DNA fragment was cloned by PCR with a pair of primers designed according to the vector sequence of the cDNA library and sequenced by Agilent Technologies. The following primers for genomic PCR cloning were used: forward, 5′-CAACCTTTAACGTCGGATGG-3′, and reverse, 5′-GGCAGGAACTGCTTACCACA-3′.

**A receptor array assay**. Plasmids of different human transmembrane genes were transfected with Lipofectamine 2000 into 293T cells in 384-well plates. Eight hours after transfection, 50 ng SEMA4A-Fc and 50 ng anti-human Ig FMAT blue secondary antibody were added into each well. The plates were read 24 h after transfection by the Applied Biosystems 8200 cellular detection system and analyzed by CDS 8200 software.

**Binding assay of SEMA4A and ILT-4 by FACS**. ILT-4-expressing cell line or activated human CD4+T cells was stained with APC-anti-ILT-4 mAb (42D1, Bioledgend; 287219, R&D systems) or APC-SEAM4AFc (R&D system), with or without SEMA4A-Fc fusion protein (R&D System), or hIgG as control. Conversely, SEMA4A-expressing cell line was stained with APC-anti-SEMA4A mAb (house hold) or APC-ILT-4-Fc (Zenon labeling kit, Life Technologies), with or without ILT-4-Fc fusion protein (R&D System) or anti-SEMA4A mAb, and hIgG or mIgG were used as controls. The binding signals were detected by FACS.

**Surface plasmon resonance**. The binding affinity between hSEMA4A and hILT-4 were carried out by surface plasmon resonance at 25 °C on a BIAcore T100 machine (Biacore AB). Recombinant SEMA4A-Fc (Sino Bio. Inc, 25 µg ml⁻¹) was coated on the series S sensor CM5 chip (GE Healthcare Life Science). ILT-4-Fc (R&D system) was diluted from 10 to 0.625 µM at ×2 fold ratio and used as follow through.

**Western blot analysis**. Freshly purified naive or pre-activated CD4+T cells were lysed with sample buffer, and then all the lysate were loaded on Novex 4–12% Bis-Tris Gels (Life Technologies) following blotted with Abs: anti-STAT6 (D3H4, 1:1000 dilution), anti-phosphorylated STAT6 (Tyr641, 1:1000 dilution), and anti-β Actin (8H10D10, 1:1000 dilution) all from Cell Signaling.

**Immunohistology analysis**. Frozen sections of human tonsil, normal, and asthma lung tissue (Biochain) were blocked with 10% goat serum and then stained with anti-SEMA4A mAb (house hold, 1:200 dilution) at room temperature for 2 h. After washing, Horseradish Peroxidase (HRP)-conjugated goat anti-mouse antibody (ab19194, 1:2000 dilution) from Abcam was applied to stain for 15 min. The slides were washed and incubated with substrate DAB Chromogen (D7304, Sigma). Finally, Haematoxylin (H3136, Sigma) was applied to counter stain according to the manufacture's instruction. For double immunofluorescence staining, the slides were stained with anti-SEMA4A mAb, followed by Alexa Fluor 594-labeled anti-mIgG antibody (A-11020, 1:500 dilution) from Thermo Fisher Scientific. Meanwhile, anti-CD11c Ab (NB110-40766, 1;200 dilution) from Novus Biologicals and Alexa Fluor 488-labeled Goat Anti-Rabbit IgG H&L Abs (ab150077, 1:500 dilution) from Abcam were applied for staining. Images were acquired by using an inverted microscope, BX41 (Olympus, Tokyo).

**Confocal microscopy**. Frozen asthma lung tissue slides (BioChain Institute Inc., T1236152Ld-1) were fixed in 4% paraformaldehyde for 20 min and permeabilized with 0.1% Triton X-100 for 5 min, then blocked for 30 min with 5% BSA, incubated with primary antibodies Rat anti-CD4 mAb (ab34276, 1:200 dilution) from Abcam, Rabbit anti-LILRB2 (ILT-4) polyclonal antibody (PAB153Hu01, 1:200 dilution) from Cloud-Clone Corp, and Cy5 conjugated mouse anti-CD3 antibody (LS-C351533, 1:100 dilution) from LifeSpan BioSciences for 2 h, followed by Alexa Fluor 488 goat anti-rabbit secondary antibody (ab150077, 1:1000) from Abcam and Cy3 conjugated donkey anti-Rat secondary antibody (AP189C, 1:1000) from EMD Millipore for 1 h, stained with DAPI (4′,6-diamidino-2-phenylindole) for 5 min and then examined with Nikon A1 laser confocal microscope (Nikon, Tokyo, Japan). Images of cells were quantified with Nikon Confocal Software.

**Statistical analysis**. The data are presented as mean value ± standard deviation (S.D.) and analyzed using Student's T-tests. Statistical significance is expressed as follows: *p < 0.05, **p < 0.01, ***p < 0.001, ****p < 0.0001, and NS indicates not significant.

**Data availability**. Microarray data that support the findings of this study have been deposited in the Gene Expression Omnibus with the primary accession codes GSE109348. All other data supporting the findings of this study are available within the article and its Supplementary Information files are available from the corresponding author on reasonable request.

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

# ARTICLE

26. Wang, Y. H. et al. IL-25 augments type 2 immune responses by enhancing the expansion and functions of TSLP-DC-activated Th2 memory cells. *J. Exp. Med.* **204**, 1837–1847 (2007).

27. Lu, N. et al. TSLP and IL-7 use two different mechanisms to regulate human CD4$^+$ T cell homeostasis. *J. Exp. Med.* **206**, 2111–2119 (2009).

28. Szabo, S. J. et al. A novel transcription factor, T-bet, directs Th1 lineage commitment. *Cell* **100**, 655–669 (2000).

29. Yamashita, M. et al. Essential role of GATA3 for the maintenance of type 2 helper T (Th2) cytokine production and chromatin remodeling at the Th2 cytokine gene loci. *J. Biol. Chem.* **279**, 26983–26990 (2004).

30. Zhu, J. et al. Conditional deletion of GATA3 shows its essential function in T (H)1–T(H)2 responses. *Nat. Immunol.* **5**, 1157–1165 (2004).

31. Hou, J. et al. An interleukin-4-induced transcription factor: IL-4Stat. *Science* **265**, 1701–1706 (1994).

32. McKenzie, A. N. Regulation of T helper type 2 cell immunity by interleukin-4 and interleukin-13. *Pharmacol. Ther.* **88**, 143–151 (2000).

33. Mann, D. L. et al. Origin of the HIV-susceptible human CD4$^+$ cell line H9. *AIDS Res. Hum. Retroviruses* **5**, 253–255 (1989).

34. Yao, S. et al. B7-h2 is a costimulatory ligand for CD28 in human. *Immunity* **34**, 729–740 (2011).

35. Battaglia, A. et al. Neuropilin-1 expression identifies a subset of regulatory T cells in human lymph nodes that is modulated by preoperative chemoradiation therapy in cervical cancer. *Immunology* **123**, 129–138 (2008).

36. Milpied, P. et al. Neuropilin-1 is not a marker of human Foxp3$^+$ Treg. *Eur. J. Immunol.* **39**, 1466–1471 (2009).

37. Battaglia, A. et al. Metastatic tumour cells favour the generation of a tolerogenic milieu in tumour draining lymph node in patients with early cervical cancer. *Cancer Immunol. Immunother.* **58**, 1363–1373 (2009).

38. E, X. Q. et al. Distribution of regulatory T cells and interaction with dendritic cells in the synovium of rheumatoid arthritis. *Scand. J. Rheumatol.* **41**, 413–420 (2012).

39. Yadav, M., Stephan, S. & Bluestone, J. A. Peripherally induced tregs – role in immune homeostasis and autoimmunity. *Front. Immunol.* **4**, 232 (2013).

40. Piechnik, A. et al. The VEGF receptor, neuropilin-1, represents a promising novel target for chronic lymphocytic leukemia patients. *Int. J. Cancer* **133**, 1489–1496 (2013).

41. Rennert, P. D. et al. T cell, Ig domain, mucin domain-2 gene-deficient mice reveal a novel mechanism for the regulation of Th2 immune responses and airway inflammation. *J. Immunol.* **177**, 4311–4321 (2006).

42. Brown, D., Trowsdale, J. & Allen, R. The LILR family: modulators of innate and adaptive immune pathways in health and disease. *Tissue Antigens* **64**, 215–225 (2004).

43. Chang, C. C. et al. Tolerization of dendritic cells by T(S) cells: the crucial role of inhibitory receptors ILT3 and ILT4. *Nat. Immunol.* **3**, 237–243 (2002).

44. Colonna, M. et al. Human myelomonocytic cells express an inhibitory receptor for classical and nonclassical MHC class I molecules. *J. Immunol.* **160**, 3096–3100 (1998).

45. Shiroishi, M. et al. Human inhibitory receptors Ig-like transcript 2 (ILT2) and ILT4 compete with CD8 for MHC class I binding and bind preferentially to HLA-G. *Proc. Natl Acad. Sci. USA* **100**, 8856–8861 (2003).

46. Shiroishi, M. et al. Structural basis for recognition of the nonclassical MHC molecule HLA-G by the leukocyte Ig-like receptor B2 (LILRB2/LIR2/ILT4/CD85d). *Proc. Natl Acad. Sci. USA* **103**, 16412–16417 (2006).

47. Liang, S. et al. Modulation of dendritic cell differentiation by HLA-G and ILT4 requires the IL-6–STAT3 signaling pathway. *Proc. Natl Acad. Sci. USA* **105**, 8357–8362 (2008).

48. Lichterfeld, M. et al. A viral CTL escape mutation leading to immunoglobulin-like transcript 4-mediated functional inhibition of myelomonocytic cells. *J. Exp. Med.* **204**, 2813–2824 (2007).

49. Hofer, J. et al. Ig-like transcript 4 as a cellular receptor for soluble complement fragment C4d. *FASEB J.* **30**, 1492–1503 (2016).

## Acknowledgements

We thank Dr. Yong-jun Liu supported, supervised the project and revised manuscript kindly. We thank the M.D. Anderson Cancer Center core facility for the use of the flowcytometer for cell sorting, the pathology facility for the histological staining and the mAb facility for antibody generation.

## Author contributions

N.L. developed the concept, designed, and performed the experiments, analyzed the data, prepared the figures and wrote the manuscript; Y.L. analyzed the data, prepared the figures, and wrote the manuscript; J.X. performed Confocal microscopy, Z.Z. designed, analyzed and supervised the ILT-4 blocking studies; Y.S. provided the human asthmatic lung tissue biopsy samples; S.Y. and L.P.C. performed the receptor membrane molecule array analysis.
