## [Peer Review File · Nature Communications]

Reviewers' comments:

Reviewer #1 (Sema4a, lupus)(Remarks to the Author):

The issue of semaphorins being highly important in a wide spectrum of immune responses is a hot issue. Many semas such as sema3A and sema4A are inhibitory/stimulatory but do not have always a specific receptor or at least many of the receptors are still not well identified. This is why, the authors did a very nice step in elegantly showed the receptor ILT-4 on Th2 cells. The methodology and discussion are well designed and of high level, and therefore convincing. Some minor comments:

1. Why clean/ purified human sema4A was not applied and instead the authors chose a complicated sema4a-L expressing cells? A purified sema would allow us to see a calibration of doses and spectrum of responses!!
2. Is bound or soluble sema4A do the same. What sheded sema4A does in vivo? Is it known??
3. Can sema4A stimulate B cells and what is the relevance of this?? more IgE or IL-4?
4. Sema4A stimulates Treg cells. What is the relevance of this when we discuss the Th2 stimulation?

Discussion should include clinical relevance of these findings

All together this is a nice paper to read

Reviewer #2 (Airway inflammation, Th function)(Remarks to the Author):

Semaphorin-4A (Sema4A)-Tim-2 axis has been implicated as a co-stimulation pathway in mouse T cells. Sema4A promotes Th1 immune responses by binding to Tim-2 in mice. However, it remains unclear if Sema4A acts as a co-stimulatory signal on human T cells as well, since Tim-2 is not conserved in the human genome. In this manuscript, the authors have shown that Sema4A is expressed in various human antigen presenting cells and co-stimulates human CD4+ T cell while promoting Th2 cytokine production. The co-stimulation effect of Sema4A is dependent on ILT-4 expressed on activated human T cells. The different signaling properties of Tim-2 and ILT-4 may explain the different co-stimulatory effect of Sema4A on human and mouse T cells-this making their report both novel and an important advance for the general immunology community.

Comments/concerns:

1. The authors have shown that Sema4A expression is elevated in asthma patient lung tissues. It would be more relevant to know if ILT-4 is expressed on T cells isolated from asthmatic lung tissues, or at least ILT-4 staining co-localizes in human asthmatic lung to CD4 T cells.
2. The authors imply in the abstract that asthma is an autoimmune disease-but there are no solid data suggesting that this is true and the reference should be deleted.
3. Fig. 2. Only representative data are shown. Aggregate data including statistical analyses are needed.
4. Fig. 3. Activation of STAT6-panel E. How do the investigators believe this is occurring-is this directly through activation of Sema4a or indirectly through the IL-4/13 that the T cells are producing as documented in panel C? If the latter, the result is trivial and can be excluded-this result is highly predictable and adds nothing to the study. A far more interesting result-and very important to clarify for this study- would be to show if Sema4a activation induces STAT6 activation independently IL-4/IL-13/IL-4R α . This could be easily tested by culturing naïve T cells with anti-CD3 and Sema4a-expressing L cells while blocking IL-4R α -then determine if you still get a Th2 phenotype and STAT6 activation.
5. Discussion. The discussion reasonably places prior work on Sem4A in the context of the current data. Please further discuss the current findings regarding prior work with ILT-4, including binding to fragments of C4(1).
6. General. The font size used in some of the flow plots is unreadable. Please increase font size to

readable proportions. The manuscript further contains many grammatical mistakes and misspellings that should be corrected with an English editor.

References

1. Hofer J, Forster F, Isenman DE, et al. Ig-like transcript 4 as a cellular receptor for soluble complement fragment C4d. *FASEB J* 2016;30:1492-503.

David B. Corry, M.D.

Wen Lu, PhD

Oct 30, 2017

Dear Editor,

Thank you very much for your taking care of our manuscript entitled "Human Semaphorin-4A drives exuberant Th2 responses by binding to receptor ILT-4" (NCOMMS-17-15409A). Also we'd like to thank all the reviewers to have read and made comments to our manuscript. Based on their advice, we have performed further experiments and revised the manuscript. Here are the details of a point by point reply to the questions:

Reviewer #1 (Sema4a, lupus)(Remarks to the Author):

The issue of semaphorins being highly important in a wide spectrum of immune responses is a hot issue. Many semas such as sema3A and sema4A are inhibitory/stimulatory but do not have always a specific receptor or at least many of the receptors are still not well identified. This is why, the authors did a very nice step in elegantly showed the receptor ILT-4 on Th2 cells. The methodology and discussion are well designed and of high level, and therefore convincing. Some minor comments:

1. Why clean/ purified human sema4A was not applied and instead the authors chose a complicated sema4a-L expressing cells? A purified sema would allow

us to see a calibration of doses and spectrum of responses!!

Reply:

Thank you for the comment.

In our study, we chose both purified Sema4A-Fc fusion protein and Sema4A-L expressing cells to show the same biological function of Sema4A as indicated in Fig 2 and Fig 6. In some experiments, L cell system was applied due to its stable expression level of Sema4A and CD32 to avoid different batch of purified protein to give different background. And also, with CD32 co-expressing, very low dose of OKT3 (0.2ug/ml) coating can maintain CD4+ T cells get marginally activated and differentiate into Th2 cells.

2. Is bound or soluble sema4A do the same. What sheded sema4A does in vivo? Is it known??

Reply:

Thank you for the comment.

In our current culture system, only bound Sema4A can enhance CD4+ naïve T cells proliferation and Th2 differentiation.

In the previous study, Morihana et al. found that in Tim2^{-/-} mice, a systemic administration of Sema4A-Fc fusion protein can significantly down-regulate the allergic response. It suggested that Sema4A exerts its inhibitory activity on allergic airway inflammation through its other functional receptor(s) instead of Tim-2¹¹. In human, Wang et al recently disclosed that increased levels of

Sema4A was found in synovial tissues, fluids, and sera of RA patients. It also correlates with Disease Activity Score (DAS)¹². Nakatsuji Y et al. found that in patients with MS (Multiple Sclerosis), serum Sema4A shed in a subgroup of the patient with significantly higher proportion of IL-17-producing CD4+T cells, higher IL-2 levels, resistance to first-line IFN- β therapy¹³.

So far, soluble Sema4A function in human is further needed to be explored.

3. Can sema4A stimulate B cells and what is the relevance of this?? more IgE or IL-4?

Reply:

Thank you for the comment.

So far, we have not explored the mechanism of Sema4A regulation on human B cells. It will be a very interesting potential project in the future.

In mouse model, it was observed that in allergen-treated Sema4A^{-/-} mice, local/systemic Th2 cytokine IL-13 production, sera allergen specific IgG1/IgG2b/IgE levels were increased, and the levels of Treg cells were decreased^{9,14}.

4. Sema4A stimulates Treg cells. What is the relevance of this when we discuss the Th2 stimulation? Discussion should include clinical relevance of these findings.

Reply:

Thank you for the comment.

In Sema4A^{-/-} mouse model of OVA-specific experimental asthma, Bronchoalveolar Lavage (BAL) fluid contains elevated levels of Th2 cytokines and IgE, as well as higher levels of Pulmonary eosinophil infiltration, however the levels of Treg cells were decreased^{9,14}. In mouse model, Sema4A is required for the function and stability of regulatory T (Treg) cells by binding to Treg-cell-expressed receptor neuropilin-1 (Nrp1)^{21,22}. Mouse Neuropilin 1 acts as a co-receptor for a number of extracellular ligands : class III and class IV semaphorins (SEMA3A/SEMA4A, respectively). Similar to SEMA 3A, SEMA4A binds to the CUB domain of NRP1, but at a lower affinity than SEMA3A^{23,24}. Studies into NRP1 expression on human Tregs demonstrate significant differences between the patterns of NRP1 expression in humans and mice. In human, higher NRP1 expression was observed on CD4⁺ T cells isolated from secondary lymphoid tissue compared with peripheral blood^{38,39,40}. One study observed significant NRP1 expression on Tregs isolated from the synovial fluid of rheumatoid arthritis patients⁴¹. Bluestone's lab also described NRP1⁺ Treg accumulation in inflamed muscles⁴². In addition, there is one recent study reported that NRP1 was significantly upregulated on Tregs isolated from the peripheral blood of chronic lymphocytic leukaemia (CLL) patients⁴³. Battaglia et al. also reported that Tregs isolated from metastatic tumour draining lymph nodes (TDLN) were significantly more enriched for NRP1 than metastasis-free TDLN^{38,40}. It seems that the exact nature and function of NRP1 expression in

humans, both under homeostatic and inflammatory conditions, still remains to be elucidated.

In our current study, when the CD4+ naïve T cells were isolated, CD4+CD25+^{high} T cells (Treg) cells were excluded. Then, with Th2/Th1 stimulation (instead of Treg stimulation), proliferation and differentiation assay of these purified CD4+ T cell were applied. Finally, we found that human Sema4A can co-stimulate CD4+ T cells and enhance Th2 differentiation through its receptor ILT-4 on activated T cells.

We have added relevance of these findings in Discussion.

Reviewer #2 (Airway inflammation, Th function)(Remarks to the Author):

Semaphorin-4A (Sema4A)-Tim-2 axis has been implicated as a co-stimulation pathway in mouse T cells. Sema4A promotes Th1 immune responses by binding to Tim-2 in mice. However, it remains unclear if Sema4A acts as a co-stimulatory signal on human T cells as well, since Tim-2 is not conserved in the human genome. In this manuscript, the authors have shown that Sema4A is expressed in various human antigen presenting cells and co-stimulates human CD4+ T cell while promoting Th2 cytokine production. The co-stimulation effect of Sema4A is dependent on ILT-4 expressed on activated human T cells. The different signaling properties of Tim-2 and ILT-4 may

explain the different co-stimulatory effect of Sema4A on human and mouse T cells-this making their report both novel and an important advance for the general immunology community.

Comments/concerns:

1. The authors have shown that Sema4A expression is elevated in asthma patient lung tissues. It would be more relevant to know if ILT-4 is expressed on T cells isolated from asthmatic lung tissues, or at least ILT-4 staining co-localizes in human asthmatic lung to CD4 T cells.

Reply:

Thank you for the comment.

We have provided new data showing that ILT-4 colocalizes with CD4 T cells in asthmatic lung tissues as indicated in Supplementary Figure 5 in the revised manuscript.

2. The authors imply in the abstract that asthma is an autoimmune disease-but there are no solid data suggesting that this is true and the reference should be deleted.

Reply:

Thank you for the comment.

We have removed this sentence and the reference.

3. Fig. 2. Only representative data are shown. Aggregate data including statistical analyses are needed.

Reply:

Thank you for the comment.

In our study, human CD4+ naïve T cells were purified from different donors. The capacity of T cells proliferation in different donor is quite different. Even the cells were harvested at the same day, as show in figure 2, the percentage of proliferated cells is quite different. However, the trend is always the same.

4. Fig. 3. Activation of STAT6-panel E. How do the investigators believe this is occurring-is this directly through activation of Sema4a or indirectly through the IL-4/13 that the T cells are producing as documented in panel C? If the latter, the result is trivial and can be excluded-this result is highly predictable and adds nothing to the study. A far more interesting result-and very important to clarify for this study- would be to show if Sema4a activation induces STAT6 activation independently IL-4/IL-13/IL-4R α . This could be easily tested by culturing naïve T cells with anti-CD3 and Sema4a-expressing L cells while blocking IL-4R α -then determine if you still get a Th2 phenotype and STAT6 activation.

Reply:

Thank you for the comment.

We have provided new data showing that Sema4A regulate STAT6 activation

on pre-stimulated CD4+ T cells as indicated in Supplementary Figure 6 in the revised manuscript.

5. Discussion. The discussion reasonably places prior work on Sem4A in the context of the current data. Please further discuss the current findings regarding prior work with ILT-4, including binding to fragments of C4(1).

Ref: Hofer J, Forster F, Isenman DE, et al. Ig-like transcript 4 as a cellular receptor for soluble complement fragment C4d. FASEB J 2016;30:1492-503.

Reply:

Thank you for the comment.

Yes, we added ILT-4 binding of fragments of C4 in discussion.

6. General. The font size used in some of the flow plots is unreadable. Please increase font size to readable proportions. The manuscript further contains may grammatical mistakes and misspellings that should be corrected with an English editor.

Reply:

Thank you for the comment.

We increased font size in the figures and we corrected mistakes carefully.

Sincerely,

Ning Lu

REVIEWERS' COMMENTS:

Reviewer #1 (Remarks to the Author):

All comments are replied properly and I have no more comments

Reviewer #2 (Remarks to the Author):

In the revised manuscript, the authors have thoroughly addressed reviewers' comments and provided new data. Supplementary figure 5 shows most lung infiltrating CD4 T cells are ILT-4 positive in patients' asthmatic lung tissue, which further support the clinical relevance of Sema4A-ILT-4 axis in human diseases. Supplementary figure 6 shows Sema4A enhances STAT6 phosphorylation in Th0 cells independent of IL-4, which proposes a new pathway regulating T helper cell differentiation.

Minor comments:

Fig. 2. Only representative data are shown. Aggregate data including statistical analyses are needed. – There are statistical analysis methods to analyze non-normally distributed matched samples, such as wilcoxon signed-rank test. It is critical to provide aggregate data including proper statistical analyses to show if Sema4A costimulates T cell proliferation. If "the trend is always the same" for most donor cells, aggregate data will provide a highly statistic significant result.

Jan 5, 2018

Dear Editor,

Thank you very much for taking care of our manuscript (NCOMMS-17-15409A) entitled "Human Semaphorin-4A drives exuberant Th2 responses by binding to receptor ILT-4" revision. We are happy to see that one of the reviewers was satisfied with our answers to his questions addressed in the last round revision. For the questions raised from the other reviewer, here are the details of a point by point reply:

Reviewer #1 (Remarks to the Author):

All comments are replied properly and I have no more comments

Reviewer #2 (Remarks to the Author):

In the revised manuscript, the authors have thoroughly addressed reviewers' comments and provided new data. Supplementary figure 5 shows most lung infiltrating CD4 T cells are ILT-4 positive in patients' asthmatic lung tissue, which further support the clinical relevance of Sema4A-ILT-4 axis in human diseases. Supplementary figure 6 shows Sema4A enhances STAT6 phosphorylation in Th0 cells independent of IL-4, which proposes a new pathway regulating T helper cell differentiation.

Minor comments:

Fig. 2. Only representative data are shown. Aggregate data including statistical

analyses are needed. – There are statistical analysis methods to analyze non-normally distributed matched samples, such as wilcoxon signed-rank test. It is critical to provide aggregate data including proper statistical analyses to show if Sema4A costimulates T cell proliferation. If “the trend is always the same” for most donor cells, aggregate data will provide a highly statistic significant result.

Reply:

Thank you for the comment.

In Fig 2, we only posted representative data. Actually, for each condition, we had more than 3 different donors’ data. As analyzed the aggregate data as follows, we found that, there is significant difference between each treatment in each group.

Group 1 (Fig 2a left up)		Group 2 (Fig2a left bottom)	
antiCD3+hIgG	antiCD3+SEMA4AFc	aCD3	aCD3+SEMA4AFc
10.30%	62%	34%	65%
9.50%	85.60%	32%	68.20%
11.70%	91.30%	30.50%	68.80%

Group 3 (Fig 2b)		Group 4 (Fig2c)	
mIgG	antiSEMA4A	mIgG	antiSEMA4A
70.20%	15.60%	89.80%	6.65%
68%	17.60%	89.40%	7.75%
64.80%	20.10%	90.70%	4%

	Group 1	Group 2	Group 3	Group 4
P value of Student T test	0.01603	0.0000272	0.0000218	0.0000372

Our data shows that human SEMA4A strongly co-stimulates T cells proliferation.

Sincerely,

Ning Lu